# A small-molecule inhibitor of lectin-like oxidized LDL receptor-1 acts by stabilizing an inactive receptor tetramer state

Gisela Schnapp[1], Heike Neubauer[1], Frank H. Büttner[1], Sandra Handschuh[1], Iain Lingard [1,3], Ralf Heilker[1], Klaus Klinder[1], Jürgen Prestle[1], Rainer Walter[1], Michael Wolff[1], Markus Zeeb [1], Francois Debaene[2], Herbert Nar [1] & Dennis Fiegen [1✉]

The C-type lectin family member lectin-like oxidized LDL receptor-1 (LOX-1) has been object of intensive research. Its modulation may offer a broad spectrum of therapeutic interventions ranging from cardiovascular diseases to cancer. LOX-1 mediates uptake of oxLDL by vascular cells and plays an important role in the initiation of endothelial dysfunction and its progression to atherosclerosis. So far only a few compounds targeting oxLDL-LOX-1 interaction are reported with a limited level of characterization. Here we describe the identification and characterization of BI-0115, a selective small molecule inhibitor of LOX-1 that blocks cellular uptake of oxLDL. Identified by a high throughput screening campaign, biophysical analysis shows that BI-0115 binding triggers receptor inhibition by formation of dimers of the homodimeric ligand binding domain. The structure of LOX-1 bound to BI-0115 shows that inter-ligand interactions at the receptor interfaces are key to the formation of the receptor tetramer thereby blocking oxLDL binding.

[1] Boehringer Ingelheim Pharma GmbH & Co. KG, 88397 Biberach, Germany. [2] NovAliX, BioParc, 850 bld Sebastien Brant, 67400 Illkirch, France. [3] Present address: Aptuit (Verona) Srl, an Evotec Company, Via Alessandro Fleming, 4, 37135 Verona, Italy. ✉email: dennis.fiegen@boehringer-ingelheim.com

A therosclerosis is a chronic inflammatory vascular disease and remains the leading cause of mortality and morbidity in industrialized countries. Besides the major risk factors such as high serum levels of total cholesterol or low-density lipoprotein cholesterol (LDLC), non-traditional risk factors are emerging as being equally important for predisposing to this disease, as many coronary artery disease (CAD) events are not prevented by only LDLC lowering therapy. Among these, elevated plasma levels of oxidized low-density lipoproteins (oxLDL) have been described to play a role in many pro-atherogenic processes, from plaque formation to plaque destabilization[1]. Lectin-like oxLDL receptor 1 (LOX-1) was discovered and described in 1997[2] as the major cell surface receptor for oxLDL in endothelial cells, macrophages, platelets and smooth muscle cells. It is a type II integral membrane glycoprotein and consists of a short N-terminal cytoplasmic portion, a transmembrane domain, a connecting neck domain regulating receptor oligomerization, and an extracellular C-type lectin-like domain (CTLD) at the C-terminus, involved in ligand binding[3]. Its expression, being almost undetectable under normal physiological conditions, is induced several fold in vascular endothelium of human atherosclerosis[4], hypertension[5] and myocardial ischemia[6]. LOX-1 binding to oxLDL leads to its internalization and proteolytic degradation. Activation of LOX-1 by oxLDL was shown to stimulate adhesion molecule and pro-angiogenic protein expression, pro-inflammatory signaling pathways and thus promotes oxidative stress, inflammation, endothelial dysfunction and apoptosis within the arterial vessel wall and atherosclerotic plaque formation and progression[3,7–9]. The key role of LOX-1 in the pathobiology of atherosclerosis has been confirmed through gene knockout and overexpression in animal models together with antibody treatment data[10–12].

LOX-1 can be proteolytically cleaved with the soluble form (sLOX-1) being released into the circulation[13]. Elevated sLOX-1 levels in the circulation were significantly associated with the incidence of coronary artery disease and ischemic stroke and were therefore proposed as useful diagnostic and prognostic biomarkers for evaluating the state and risk of atherosclerosis and atherosclerosis-related diseases (for reviews see refs. [8,14–16]). Furthermore, some human genetic linkage studies implicate LOX-1 polymorphisms in cardiovascular disease (CVD) susceptibility[17,18]. Interestingly, some recent studies showed upregulated expression in several types of cancer and positive correlation with tumor stage and grade. This suggests that LOX-1 may also play a critical role in tumorigenesis and has the potential as a biomarker in oncology[19–22].

These data suggest that blocking LOX-1 function might be an attractive therapeutic concept for atherosclerosis and associated vascular diseases, but may also have applications beyond those diseases. However, therapeutic approaches, interfering with LOX-1 function are only starting to evolve. A recent study[23] identified a few potential small molecule inhibitors of LOX-1 using virtual screening techniques, but mechanistic data are missing.

The crystal structures of the human LOX-1 receptor showed that it forms a heart-shaped homodimer[24–26]. The three intra-chain disulfide bridges stabilize the overall fold of the subunits, whereas the inter-chain disulfide bridge at position 140 results in dimer formation. A hydrophobic tunnel is present at the homodimer interface. In one study a dioxane molecule from the crystallization solution occupied this space[24]. Nakano et al.[26] demonstrated that residues surrounding this tunnel are important in the self-assembly of the canonical dimer. Oligomerization of the dimeric receptor on the cell surface is important for LOX-1 function. It is suggested that at least three dimeric LOX-1 receptors bind to oxLDL[27,28]. Especially important for the

binding of the negatively charged oxLDL is the basic spine structure, exposing three arginine residues (Arg208, Arg229 and Arg248) on the predicted binding surface[25].

Here we describe the identification of small molecules, which potently block cellular uptake of fluorescently labelled human oxLDL in a high throughput assay. Using a set of counter screens and a variety of biophysical methods we demonstrate specific receptor binding and elucidate the mode-of-action of the active compounds. Finally, we provide the crystallographic structure of a small molecule (BI-0115)-LOX-1 receptor complex, which shows atomic details of the protein-ligand interaction and the structural basis of the inhibitory mode-of-action.

## Results

**Small molecules blocking LOX-1 mediated oxLDL uptake**. To identify small molecule inhibitors of LOX-1 mediated oxLDL uptake we developed a cellular high throughput screening (HTS) assay. Small molecule compounds were screened for their ability to block the uptake of AlexaFluor594 (AF594)-labeled human oxLDL into a CHO-K1 cell line with inducible expression of the human LOX-1 receptor. This readout was visualized and images were quantified by fluorescence microscopy (Fig. 1a). As positive controls, cells without induction of LOX-1 as well as a blocking anti-LOX-1 antibody[29] was used (Fig. 1b). More than one million different compounds were screened. Lead-like compounds (molecular weight above 270 Da) were tested at a concentration of 5 μg/ml, whereas a fragment subset (molecular weight below 270 Da) was tested at a tenfold higher concentration to ensure detection of more weakly active but ligand-efficient fragment-like compounds. Employing a hit criterion of lower than 50% of control, a hit rate of 2.8% was observed (Supplementary Fig. 1). These hits were further confirmed by performing two independent assay runs with fresh compound samples, resulting in a confirmation rate of 80% and a confirmed hit rate of 2.2% (Supplementary Fig. 2). To filter out unspecific or false-positive hits, a counter screen was developed and applied for testing the primary hits[30–32]. The assay format was identical to the primary assay besides employing a CHO cell line with inducible expression of human HDL scavenger receptor class B type I (SR-BI), an alternative scavenger receptor with low sequence and structural homology to human LOX-1 (Supplementary Fig. 1). Human SR-B1 is a cell surface glycoprotein most highly expressed in liver and steroidogenic tissues and plays a key role in mediating selective HDL cholesterol (HDL-C) uptake in the liver. Its regulation of HDL-C metabolism is believed to be relevant for cardiovascular health and the receptor was therefore chosen as a counter target[33]. Using this single assay, we eliminated compounds that blocked both receptors presumably via an unspecific, non-target related mechanism (Supplementary Fig. 3).

Employing filter criteria like < 65% control in the primary assay and a minimum difference of 30% to the SR-B1 counter screen, about 3300 compounds remained. These compounds were tested in concentration response using both assay formats (Supplementary Fig. 1). To qualify the identified compounds further we developed additional assays with the goal to identify compounds that merely quenched the AF dye fluorescence. In this assay, the fluorescence signal of human AF647- or AF594-labeled oxLDL was recorded in the presence and absence of compound. Only compounds without relevant quenching were subsequently selected for further analysis.

Finally, we obtained 11 clusters of structurally diverse compounds that showed the desired profile. One of the clusters corresponded to a tricyclic series, with BI-0115 (Fig. 1c and Table 1) being the most active representative. It exhibits an $IC_{50}$ of 5.4 μM ($n = 8$) in the LOX-1 cellular assay (Fig. 1c, Table 1 and

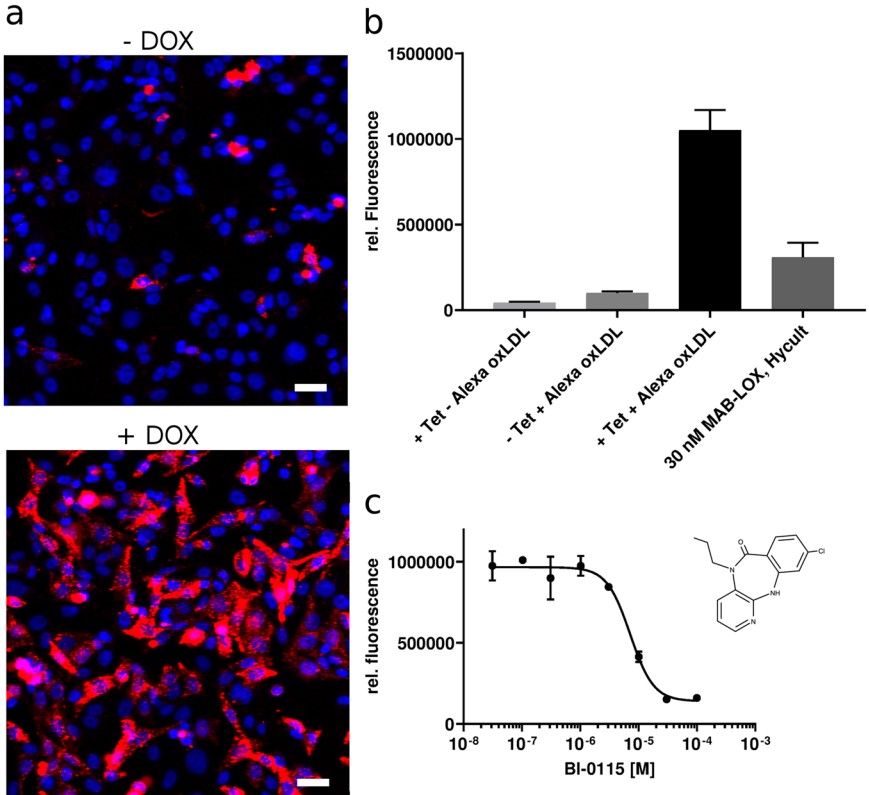

**Fig. 1 Cell based high throughput screening assay on LOX-1. a** Representative fluorescence microscopic pictures ($n = 16$) of AF647 labeled human oxLDL uptake into a CHO-K1 cell line with doxycycline inducible expression of the human LOX-1 receptor (-/+ DOX). The length of the scale bar corresponds to 20 μM. blue: Hoechst 33342, red: AF647-oxLDL **b** The HTS assay principle is based on the cellular internalization of AF594-labeled human oxLDL. AF594-oxLDL uptake in CHO-TREx-hLOX-1 is blocked by 30 nM mAb-LOX-1 (P value < 0.0001). The assay shows a window of factor 10 between background (-Tet+Alexa oxLDL) and full induction (+Tet+Alexa oxLDL). The error bars indicate the standard deviation of the measurements. **c** Representative example of a concentration response curve of BI-0115 with an $IC_{50}$ of 7.2 μM in the LOX-1 cellular uptake assay ($n = 2$ for each concentration, $R^2 = 0.98$). The chemical structure of BI-0115 is indented.

Supplementary Table 1) and no activity in the corresponding SR-B1 assay (Table 1, Supplementary Table 2).

**Biophysics confirm target interaction.** For the biophysical and structural analysis, we used two different forms of the extra-cellular part of LOX-1. LOX-1 129-273 (LOX129 in the following) contains a short fragment of the neck and the CTLD domain, whereas LOX-1 143-273 (LOX143) contains only the CTLD domain. The LOX143 construct is monomeric in solution, whereas LOX129 is dimeric due to an inter-chain disulfide bridge at Cys140[24,25].

Selected cluster representatives were analyzed by multiple biophysical methods to further validate and characterize the identified compounds. As a first method, Saturation Transfer Difference (STD)-NMR was used to demonstrate binding to the receptor. BI-0115 showed clear STD signals in the aromatic region (18.5% STD, Fig. 2a) and comparable STD effects for the methyl moiety in the aliphatic region of the $^1$H NMR spectrum (data not shown). In addition, similar STD signals of BI-0115 were observed in the presence of LOX143 suggesting that the compound binds to both LOX129 as well as LOX143 under identical conditions (each with $n = 1$).

To enable surface plasmon resonance spectroscopy (SPR), LOX129 was immobilized on a SPR chip by amine coupling and compounds were analyzed for binding. Based on the individual sensorgrams the compounds were classified as non-binders, over-stoichiometric binders or dose-dependent binders. In total four different structural classes could be confirmed as dose-dependent

and stoichiometric binders. A representative sensorgram for BI-0115 is shown in Fig. 2b. The compound shows fast binding kinetics. A steady-state analysis of the SPR data indicated a mean Kd of 4.3 μM. The mean binding ratio is 0.78 (Supplementary Table 3). Assuming a high percentage of active, binding competent protein on the chip surface, this ratio represents less than one BI-0115 molecule per LOX-1 monomer.

For additional biophysical profiling of the specific SPR binders, isothermal titration calorimetry (ITC) was used. Due to limited solubility of the compounds, an inverse setup was used by titrating LOX129 into compound solution. One representative binding isotherm of BI-0115 is shown in Fig. 2c. The mean Kd value of three independent measurements is 7 μM (Supplementary Table 4). Binding of BI-0115 is driven by a strong entropic contribution, while only relatively weak enthalpic contributions were observed. Another intriguing observation in the ITC experiments was that the molar ratio of protein and inhibitor was not 1, but 1.8, suggesting a molar ratio of protein to inhibitor of 2 (Supplementary Table 4).

Overall, biophysical characterization of the hit set was found to be crucial for the progression of the project. Following the fluorescence-quenching assay, it mainly helped to weed out false positives and confirmed a direct interaction between the target and small molecule, thereby allowing prioritization of the most promising compound series.

**BI-0115 binding induces a tetrameric co-structure.** The non-liganded LOX129 protein crystallized in a new crystal form with

**Table 1 Summary of BI-0115 and BI-1580. The chemical structures are shown at the top of the table.**

| | BI-0115 | BI-1580 |
|---|---|---|
| Calculated properties | | |
| MW | 287 | 267 |
| cLOGP | 3.6 | 2.8 |
| Target engagement and selectivity | | |
| LOX-1 IC50 [μM] | 5.4 ±1.8 | >100 |
| LE (on IC50) | 0.32 | - |
| Quench Assay [% at 30 μM] | −9.1 | 23.9 |
| SR-B1 IC50 [μM] | >172 | >100 |
| Biophysical characterization | | |
| ITC [μM]; N | 6.99 ±0.75; 1.77 ±0.34 | - |
| SPR [μM] | 4.3 ±0.4 | - |
| STD-NMR | binding | - |
| ESI-MS [LOX-1:BI-0115] | 4:2 binding | - |
| in vitro PK profile | | |
| Solubility pH4/pH7 [mg/ml] | 0.001/0.001 | - |
| micr. Stab hLM/rLM/mLM [%QH] | 77/95/95 | - |
| PAMPA permeability | $1.5 \times 10^{-6}$ | - |
| hERG IC50 [μM] | >10 | - |
| CEREP profile [at 10 μM] | clean | 72.3% 5-HT$_{2B}$ |

three crystallographically independent protomers, comprising a LOX-1 dimer in the asymmetric unit as well as a monomer positioned on a crystallographic twofold axis. The non-liganded LOX143 protein forms crystals with a CTLD dimer in the asymmetric unit (Supplementary Table 5). Both crystal forms diffract to high resolution. Structural models for LOX129 and LOX143 derived from the crystallographic data are very similar to published structures[24,25]. The oxLDL binding site, the basic spine[25] (Supplementary Fig. 6), is only partially accessible in these crystal forms. Soaking of BI-0115 into both polymorphs was not successful. Based on the convincing biophysical characterization BI-0115 was selected as the prime inhibitor candidate for co-crystallization studies. Surprisingly, the compound co-crystallized with the receptor in a new crystal form. This new form contains two LOX-1 tetramers in the asymmetric unit (Fig. 3a, Supplementary Table 5 and Supplementary Fig. 4). The two tetramers are basically identical and superpose with an r.m.s.d. of 0.45 Å. Each tetramer consists of two dimers of the known, "physiological" quaternary structure that interact with each other in a head-to-head fashion (Fig. 3a and Supplementary Fig. 5). Two BI-0115 molecules and a central water molecule, located at the interface of the two dimers, mediate the tetramerization. Supplementary Fig. 7a, b show the well-defined electron density around the two ligands. The two dimers can be transformed into each other via a nearly perfect twofold rotation axis through the central water molecule (Fig. 3a). Due to the two fold symmetry of the tetramer arrangement, BI-0115 molecules each bind to only one protomer subsite of each physiological LOX-1 dimer, while the corresponding symmetrical second site is occupied by a loop of the opposing dimer comprising residues N236, T237, Y238 and P239 (Fig. 3c). The inter-dimer contacts in the second site are primarily van der Waals interactions. An intermolecular H-bond is formed between the side chains of N255 of molecule B and Q247 of molecule D.

The symmetry in the tetramer arrangement leads to identical interactions of the two ligands, so that in the following the interactions for only one molecule are described. The binding site and the interaction between ligand and protein are mainly hydrophobic in character. Residues P201, W203, Y245, L258 and A260 line the base of the pocket. These interact with the aromatic ring and the chlorine atom of BI-0115. Important for the directionality of the interaction are two H-bonds. The aniline nitrogen that connects the two six membered rings forms a hydrogen bond to the main chain carbonyl of A259. The pyridyl nitrogen forms an H-bond to the main chain nitrogen of A259. The pyridyl ring is slightly more solvent exposed and packs against the side chain of F200 of the opposing dimer and thereby helps to connect the two dimers.

The ligand carbonyl group makes a hydrogen bond to the central water molecule (WAT in Fig. 3b) that in turn is coordinated by the side chain of Q247. This central water molecule appears to be an important mediator of interactions as it connects the two BI-0115 molecules via their carbonyl groups and the ligands with the protein (Fig. 3b). The N-propyl moiety of BI-0115 is part of the hydrophobic interface of the two BI-0115 ligands and sits in a hydrophobic pocket of the opposing dimer surrounded by residues S162, P201 and F261. The distances between closest atoms of the two BI-0115 ligands are in the range of 3.5–4.3 Å. In addition to these ligand mediated interactions between the two dimers there are inter-dimer H-bonds involved, the side chain of E254 (molecule C) interacts with the side chain of S199 (molecule A) and the side chain amide of Q247 (molecule C) with the side chain oxygen of Q247 (molecule B) (Fig. 3b).

Non-liganded and ligand-bound LOX-1 monomers have overall very similar structures with r.m.s.d. values in the range of 0.58–0.68 Å (Fig. 3e). When superposing multiple non-liganded LOX-1 dimers it became apparent that there is a large conformational plasticity in the dimeric ensemble. We defined the "ground state" as the LOX129 molecule C and its symmetry related molecule C*. They are crystallographically identical and thus related by perfect C2 symmetry (Fig. 3d). The LOX143 dimer shows only a small rotational shift, whereas the LOX129 AB dimer and the LOX143-BI-0115 AB dimer have large deviations from ideal twofold symmetry (rotation angle of only ~170°) leading to a shift of 5–6 Å of residues located at the dimer periphery (Fig. 3d). These deviations are accompanied by large conformational changes of side chains along the twofold axis. The dimer interface functions as a hinge and is build up by a hydrophobic cluster consisting of residues F261, F200, F202, F158, Y197, F190, Y156, H151 and W150, transmitting the conformational change via a sliding motion of the hydrophobic residues.

Ligand binding induces only small local conformational changes (Fig. 3e). The side chains of L258, F200, F261 and Y245 are frozen in positions also observed in non-liganded structures and thereby create space for the pyridine moiety of BI-0115. Loop binding (residues A233 to P239 of molecule C) in the symmetrical site (molecule A) also induces only small local changes. A consequence of ligand binding is thus a significant change in quaternary structure primarily by tetramer formation, which brings subunit C into a position that pushes subunit A down, a structural change based on a rotation relative to subunit B (Fig. 3a, c, d). The observed rotation angle of the complex is 170°, very similar to the maximum deviations from ideal C2 symmetry observed for the unliganded LOX-1 structures. In conclusion, compound binding to LOX-1 and the formation of the tetrameric complex occludes oxLDL from binding to the receptor by sterically blocking oxLDL interaction with the basic spine (Supplementary Fig. 6).

The overall structure activity relationships around the BI-0115 series are very steep (Supplementary Fig. 8) and can be explained

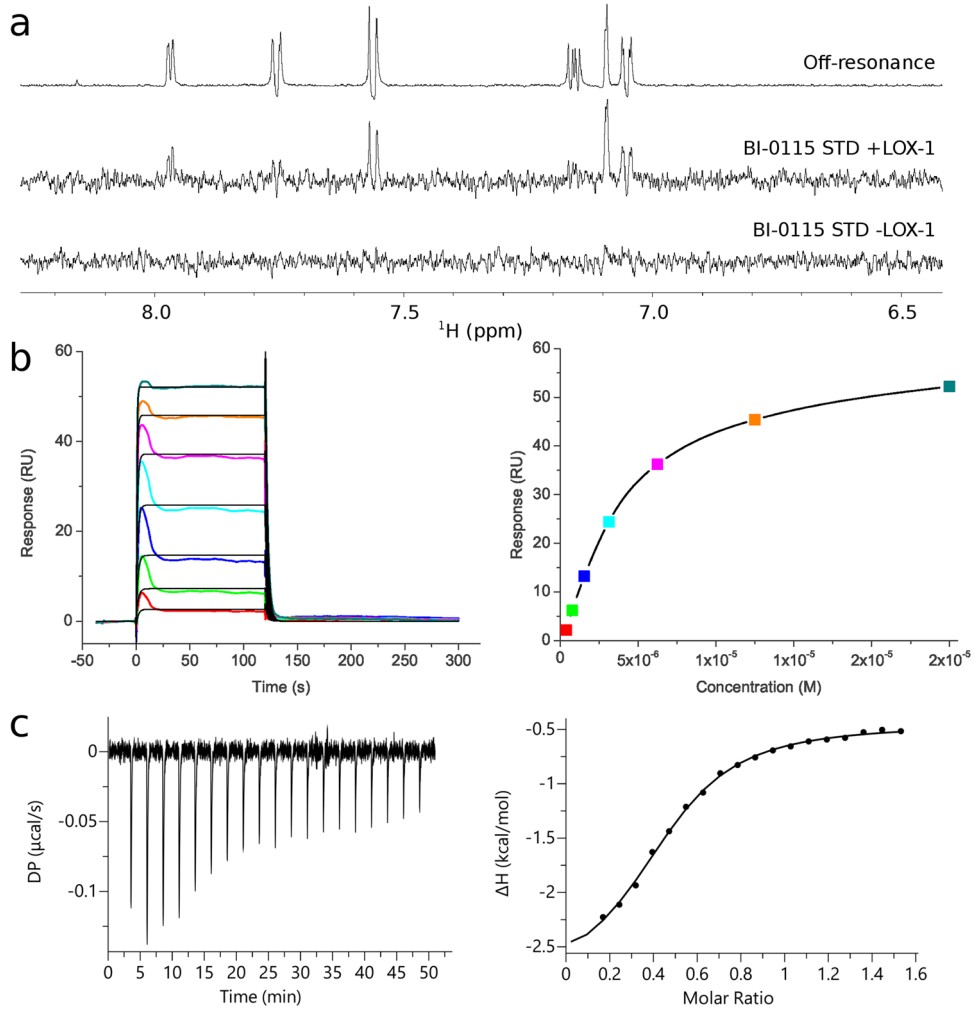

**Fig. 2 Biophysical characterization of BI-0115-LOX-1 interaction. a** Characterization of small molecule binding to LOX129 by STD NMR. Aromatic region of the off-resonance spectrum showing all expected resonances of BI-0115 (labeled Off-resonance). STD spectrum (subtraction of on- and off-resonance spectrum) displays signals at the respective resonance frequencies of the small molecule indicative for binding to LOX-1 (BI-0115 STD +LOX-1). STD spectrum of BI-0115 in the absence of LOX-1 (BI-0115 STD -LOX-1) shows no signals supporting specific binding of the compound to LOX-1 in the middle row. **b** Surface plasmon resonance binding analysis of BI-0115 to immobilized LOX129. Binding of BI-0115 was assessed by measuring dose responsive changes (from 0.391 μM to 25 μM) in the refractive index. The Kd of 4.3 μM was determined by a fit to the change of the refractive index at equilibrium ($n = 4$). The figure shows a representative example of a sensorgramm and affinity fit. **c** ITC-Titration of 300 μM LOX129 dimer into 40 μM BI-0115 compound to determine the binding affinity (mean $K_d = 6.99$ μM, $n = 3$). The left graph presents the raw data. The integrated peak areas with the corresponding fit are shown at the right.

by the available structural information. Only minor modifications are allowed in the region of the lactam nitrogen. In total only three active compounds (Supplementary Fig. 8) within this series showed activity in the cellular uptake assay. These small modifications would still fit into the narrow cavity made up by the opposing dimers. Larger or polar substitutions in this position lead to inactive compounds, as they clash with the neighboring protein molecule. Modifications in any other part of the molecule lead to inactive compounds. In the case of the negative control compound BI-1580 (Table 1), the additional methyl group at the aniline nitrogen would clash with the backbone carbonyl of A259.

**ESI-MS confirms the BI-0115-induced tetramer in solution**. The crystal structure of the LOX-1-BI-0115 complex suggests a 4:2 binding ratio. This is in agreement with the biophysical data obtained so far. The ITC measurements gave a stoichiometry of protein:ligand of 1.8 which fits to multiples of a 2:1 complex. To confirm that the crystallographically observed tetrameric arrangement exists also in solution native Electrospray-Ionization

Mass Spectrometry (ESI-MS) analysis was used. A MS experiment using covalent dimeric LOX129 under denaturing conditions was performed giving a mass of 33773 Da, corresponding to a mass difference of 13 Da to the theoretical molecular weight of 33786 Da (Supplementary Fig. 9). This mass difference corresponds to the three intramolecular and the one intermolecular disulfide bonds (14 Da). MS analysis under non denaturing conditions (Fig. 4a) measured LOX129 mainly as a dimer. Only traces of the tetrameric state were detectable. A binding experiment of BI-0115 to LOX-1 was performed using soft MS tuning parameters (Vc = 40V) in order to maintain BI-0115 interactions in the gas phase. Titration showed a dose-dependent increase of a non-covalent LOX-1 tetramer and the expected stoichiometry of protein sub-unit:ligand of 4:2 (Fig. 4b–d). Furthermore, LOX-1 showed only a low binding level of ligand with a 2:1 stoichiometry and no concentration dependent increase of the ligand bound form. The same sample analyzed at higher voltage (Vc = 60V) induced BI-0115 dissociation in the gas phase but did not impact the LOX-1 tetramer proportion. These results confirmed the tetrameric

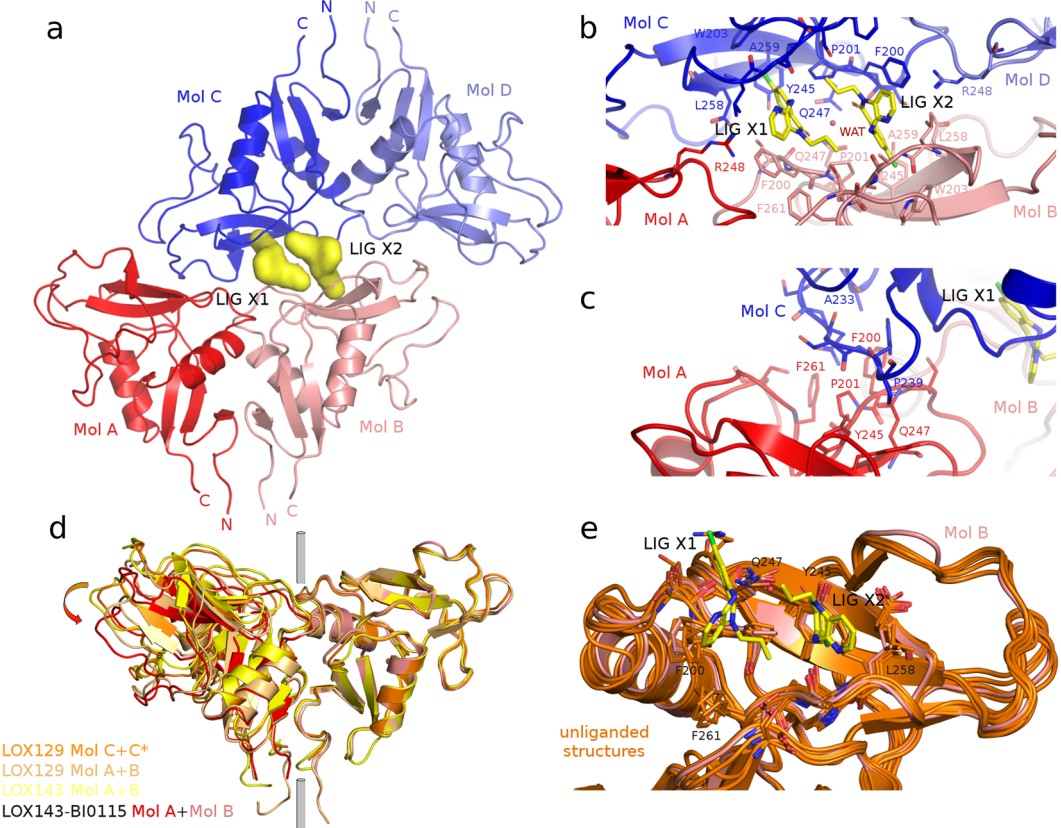

**Fig. 3 Crystal structure of the BI-0115-LOX-1 complex. a** Two molecules of BI-0115 (yellow surface, LIG X1/LIG X2) bind to two dimers of LOX-1 CTLD. One dimer is formed by molecule A (Mol A in red) and molecule B (Mol B in salmon) and the second dimer by molecule C (Mol C in blue) and molecule D (Mol D in light blue). The amino and carboxyl-termini are indicated by N and C. **b** Detailed representation of the BI-0115 - LOX-1 interaction. Amino acids surrounding the binding site are shown as lines. C atoms are colored according to the LOX-1 molecule; O and N are in red and blue, respectively. **c** Close-up view of the asymmetric non-liganded binding site. Blue colored loop A233 to P239 of molecule C binds to the hydrophobic pocket of molecule A in red. **d** Superposition of multiple LOX-1 dimers. The right LOX129 C molecule has been used as reference for the superposition. The grey cylinder indicates the twofold rotation axis. The arrow at the left hand side indicates the large shift observed in the LOX129 AB dimer and the LOX143-BI-0115 AB dimer structures. **e** Close-up view of multiple unliganded LOX-1 monomers superposed on molecule B of the LOX143-BI-0115 complex. Residues near the ligand are highlighted.

recruitment induced by the ligand (Supplementary Fig. 10) and allowed relative quantification (increase up to 33% of the 16+ charge state) of ligand induced tetramerization.

**BI-0115 is a highly selective in vitro compound**. As summarized in Table 1, BI-0115 shows a clear inhibition of oxLDL internalization with an $IC_{50}$ of 5.4 μM. To assess potential off-target liabilities, BI-0115 was tested on a panel of assays across various key pharmacological target classes (GPCRs, drug transporters, ion channels, nuclear receptors, kinases, and other non-kinase enzymes; Eurofins CEREP, France). It showed a very clean profile and therefore exhibited a very good selectivity (Supplementary Fig. 13). The control compound BI-1580 with no activity on LOX-1, tested in parallel, showed a similar selectivity profile. The most prominent activity was observed for the 5-$HT_{2B}$ agonist radio ligand with 72.3% at 10 μM (Supplementary Fig. 14). BI-0115 shows no inhibition of the hERG channel. Although the compound solubility is low, it was sufficient to get a saturable dose response in the cellular assay and the various biophysical assays. Permeability, as measured with the PAMPA assay is good. The stability in human, rat and mouse liver microsomes was rather low with in vitro clearance equivalent to 77%, 95% and 95% hepatic blood flow ($Q_H$) respectively. Based on these data,

BI-0115 is an ideal tool compound for in vitro studies to further investigate and generate understanding of the LOX-1 biology.

## Discussion

Over the last years, C-type lectin-like receptors evolved as an important and promising new family of therapeutic targets for a broad range of diseases[34]. One receptor of this large family, the lectin-like oxLDL receptor-1 (LOX-1) is the main receptor for oxLDL in endothelial cells, macrophages and vascular smooth muscle cells and upregulated during inflammation and pathological conditions[2,35]. Increased oxLDL formation and aberrant cellular uptake by scavenger receptors plays a critical role in the development of atherosclerosis. Furthermore, growing evidence supports a vital role of LOX-1, one of the scavenger receptors, at various steps of the atherosclerotic process, from endothelial dysfunction to formation and destabilization of atherosclerotic plaques[7,9,36]. Interfering with this cascade of events culminating in atherosclerosis and its sequelae, such as angina pectoris and acute coronary syndrome, is an interesting therapeutic concept. Despite this interest in discovering inhibitors for the C-type lectin-like receptors and especially LOX-1, so far only a few of limited quality are reported[23,37,38]. Five structurally unrelated LOX-1 inhibitors have been described in the literature to date[23].

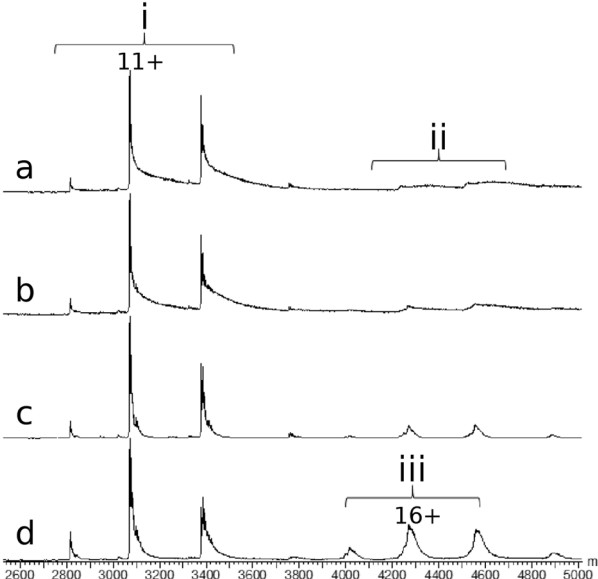

**Fig. 4 BI-0115 binding induces tetramerization in solution. a** LOX-1 tetrameric state detected at trace level without ligand. Increasing ligand concentration (**a**) 0 eq, (**b**) 2.5 eq, (**c**) 5 eq, (**d**) 10 eq induces LOX-1 tetramerization (each with $n = 1$). (i) (LOX-1)$_2$ dimer (MW: 33776 ± 1Da), (ii) (LOX-1)$_4$: 67565 ± 4Da, (iii) [(LOX-1)$_4$-BI-0115$_2$+ acetate]: 68202 ± 4 Da.

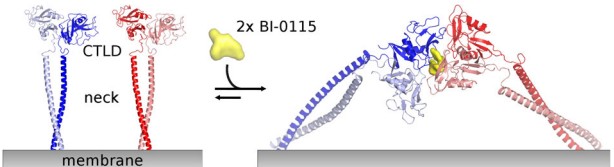

**Fig. 5 Homology model of the tetrameric complex.** The CTLD part is based on the co-crystal structure of the BI-0115-LOX-1 complex, whereas the neck domain is modeled based on the Homer coiled-coil domain PDB ID 3CVE. The grey bar at the bottom represents the membrane. One dimer is built up by one molecule colored in red and the second molecule in salmon, the second dimer by one blue and one light blue molecule. CTLD: C-type lectin-like domain.

They shift the LOX-1 melting point by 1–10 °C[23] and the most potent compound reduced oxLDL uptake with an IC$_{50}$ of around 200 nM[23]. Docking studies suggest binding to the LOX-1 tunnel[23]. Biocca et al.[38] have proposed a mechanism of statin-mediated inhibition of LOX-1, where statins bind to the same tunnel in the dimer interface and thereby stabilize LOX-1 dimers.

For the identification of compounds inhibiting the uptake of oxLDL by LOX-1, we developed a cell based fluorescence assay employing AF594-labeled oxLDL. To complement this primary screening assay, it was important to have two counter screens in place, an SR-B1 oxLDL-uptake assay and the AF fluorescence-quenching assays. Both to confirm LOX-1 selectivity and to weed out false positive compounds. In this context, the biophysical characterization of compounds by STD NMR, SPR and ITC has been crucial to confirm target interaction. The screening cascade identified a cluster of compounds that all contain a tricyclic system, with BI-0115 being the most active compound. BI-0115 efficiently reduced oxLDL internalization in the cellular context. The IC$_{50}$ values of the cellular assay (5.4 µM, Table 1) were in good agreement with the determined Kd values in SPR (4.3 µM, Table 1) and ITC (6.99 µM, Table 1).

The crystal structure of the BI-0115-LOX-1 complex was fundamental in understanding the molecular mechanism of BI-0115 inhibition. It showed that two BI-0115 molecules glue two LOX-1 dimers together, in a head-to-head fashion. This sterically blocks oxLDL access to the basic spine structure, important for high affinity oxLDL binding[25,26]. The cluster analysis revealed that the structure activity relationship for this series is quite steep. Furthermore, the observed SAR is well explained by the BI-0115-LOX-1 co-structure. The two BI-0115 molecules bind to a small pocket (Supplementary Fig. 16) that is mostly surrounded by protein residues in van-der-Waals distance. Only small modifications in the position of the propyl-moiety are tolerated. An additional level of complexity is provided by the direct neighborhood of the two BI-0115 molecules and their 2-fold symmetric

arrangement, where changes in one position might at the same time influence the affinity of the symmetry related molecule. This additionally complicates chemical optimization of this series and makes the availability of 3D structural information a prerequisite. Although SAR is quite steep, there are still growth vectors for affinity optimization, like the solvent exposed part of the pyridine ring and the n-propyl moiety. Here the SAR already shows that variations are possible (Supplementary Fig. 8).

An important confirmation for the 4:2 LOX-1:BI-0115 binding ratio, as observed in the crystal structure, has been non-denaturing mass spectrometry. This, together with Isothermal Titration Calorimetry data, helped to rule out possible alternative binding ratios (e.g. 2:2 or 2:4 LOX-1:BI-0115) and confirmed the mode of inhibition. Based on the available data, we propose that the high local concentration of the receptor on the cell surface[28] leads in the presence of BI-0115 to formation of a tetrameric state (Fig. 5). This mechanism fits to the model that C-type lectin receptors tend to oligomerize to increase their avidity for multi-valent ligands. In the case of LOX-1, receptor oligomerization is a prerequisite for high affinity oxLDL binding[27,28,36]. The additional unliganded LOX-1 structures reported here helped in mapping the conformational flexibility of the LOX-1 dimers (Fig. 3d) and demonstrated that the dimer conformation observed in the LOX-1-BI-0115 complex is very similar to unliganded LOX-1 structures. Nakano et al.[26] showed that the Phe and interfacial hydrophobic cluster is important for oxLDL binding and dimerization of LOX-1. The hydrophobic hinge at the dimer interface depicted here extends the former and could have an important biological function in flexibly adjusting to different oligomeric states.

The reaction scheme presented in Fig. 5 allows multiple reaction intermediates (Supplementary Fig. 15). One possible intermediate may be a preformed low abundancy tetrameric complex that is the result of the high local concentration of LOX-1 on the cell surface. Compound binding then stabilizes this tetrameric complex and thereby drives equilibrium to a state that cannot bind oxLDL and inhibits internalization (Supplementary Fig. 15). Alternatives are intermediates in which one LOX-1 dimer binds one or two BI-0115 molecules followed by tetramerization. Independent of the exact reaction pathway, the mode of inhibition presented here is an interesting example of the emerging concept of small molecule mediated Protein-Protein-Interaction (PPI) stabilization[39,40].

An important aspect for a target are species differences, especially when studying disease models. The sequence conservation between human, rat and mouse LOX-1 is only 66% and 60%, respectively (Supplementary Fig. 11). A prominent feature of the rat and mouse LOX-1 sequences is a ~90 amino acids insertion in

their neck domains. The sequence conservation between human and rodent CTLD domains is 68% for rat and 59% for mouse. With regard to the BI-0115 binding site, two important residues, F200 and A259, differ between the three species (Supplementary Fig. 12, Fig. 3b). F200, a serine residue in mouse and rat, makes a hydrophobic interaction to BI-0115. This interaction is not possible for the serine side chain, likely leading to a reduced activity of BI-0115 on mouse and rat LOX-1. The sequence conservation between C-type lectins is rather low. Closest paralogues to LOX-1 are CLEC7A, CLEC9A and CLEC12B with a sequence identity below 40% (Supplementary Fig. 17). As only three (F202, W203 and Y245) out of 13 residues around the BI-0115 binding site are partially conserved between these closest paralogues (Supplement Fig. 18), it is very unlikely that BI-0115 is going to bind to other C-type lectins.

The BI-0115 compound is available to the scientific community via opnMe.com, the Boehringer Ingelheim Open Innovation portal (https://opnme.com/molecules/lox-1-bi-0115) as a selective LOX-1 inhibitor for in vitro studies. We believe that this compound will be a valuable chemical probe to further study the LOX-1 biology, downstream pathways, and finally, help to elucidate its role and contribution in different pathogenic processes (like atherosclerosis or oncology) in suitable cell models.

Moreover, this new mechanism of C-type lectin-like receptor inhibition is groundbreaking as it opens new avenues to target this large and important family of receptors. It is tempting to speculate that the same inhibitory mechanism may also work for other family members and could provide a role model for C-type lectin-like receptor inhibition.

## Methods

**Cell line generation**. A recombinant CHO-K1 cell line was generated to stably express human LOX-1 under control of a Tetracycline inducible promoter: using Gateway® Cloning (Invitrogen), the hLOX-1 cDNA was cloned into pT-Rex-DEST-30 thus generating pT-Rex-EXP-30-hLOX. The CHO-K1 T-REx host cell line was then transfected with the expression vector pT-Rex-EXP-30-hLOX. Stably transfected cells were selected by Geneticin resistance. Single cell clones expressing human LOX-1 upon induction with Doxycycline were isolated and the best clone selected by qRT-PCR and FACS analysis (CHO-Trex-hLOX1 cell clone). Cells expressing human SR-BI were prepared and used for the assay in a similar fashion (CHO-Trex-hSR-B1 cell line). The obtained cells were cultured in Ham´s F-12 with L-Glutamin + 10% FCS + 10 μg/ml Blasticidin + 700 μg/ml Geneticin.

**oxLDL internalization in fluorescence microscopy**. The CHO-Trex-hLOX1 cell line was cultured on 384-well imaging-suitable collagen I-coated microtiter plates (Falcon BD) in RPMI 1640 medium with L-Glutamine (BioWhittaker, #BE12-702F), 1% Pen/Strep (Biochrom, #A2213), and 10% fetal bovine serum (FBS; Gibco, #10500-064). For induction of LOX-1 receptor expression, the cells were incubated overnight in culture medium supplemented with 0.1 μg/ml of doxycycline. In order to monitor ligand internalization, the cells were incubated with 5 nM AF647 fluorescently labeled human oxLDL for 30 min at 37 °C, 5% CO2. Then the cells were fixed for 30 min at room temperature (RT) with 4% formaldehyde (FA)/phosphate-buffered saline (PBS) solution (Sigma-Aldrich, #25,254-9). In parallel to the fixation process, the cell nuclei were stained by adding 1 μM Hoechst 33342 dye (Molecular Probes, #H-3570) to the fixation medium. All images of the respective experiments were taken using the IN Cell Analyzer 3000 (General Electric [GE] Healthcare).

**High throughput cellular oxLDL uptake assays**. Assay conditions for the CHO-Trex-hLOX1 cell line and the CHO-Trex-hSR-B1 cell line were identical. Into sterile, 384-well plates, black with clear bottom (Becton Dickinson # 359292), 50 μl of Doxycycline (final concentration of 0.1 μg/ml) prepared in cell culture medium HAM`s F-12 with L-Glutamine; (Lonza # BE12-615F) supplemented with 10% FCS (Biological Industries (# 04-001-1A)) were added to the rows 1–23. In row 24, 50 μl of cell culture medium without Doxycyclin was added followed by an incubation of these plates for 2h at 37 °C in an incubator, to allow the plates to reach an even temperature. After this incubation step, 10 μl of freshly prepared or frozen cell suspension (15.000 cells/well dissolved in cell culture medium) were added to the 384-well plates. The plates were incubated employing a humidified incubator for 18h at 37 °C/5% CO2. Cell plates were washed with a cell washer with 100 μl/well of washing buffer (HBSS1x, Gibco # 14065-072) followed by the addition of 10 μl of assay buffer HAM`s F-12 with L-Glutamine, 25 mM HEPES pH 7.2 (1M Hepes, pH 7.2 Bio Wittaker Europe #BE17-737E). After this 5 μl of test compound dissolved in assay buffer (final concentration of compound 5 μg/ml, 1% DMSO) were added to rows 1–23. For the CHO-Trex-hLOX1 cells simultaneously 5 μl of an anti-human-LOX-1 antibody (Trinova Biochem / Hycult # HM2138, final concentration 30 nM) were added to 8 wells in row 24 as an inhibiting reference control. Finally 10 μl of AF594 (Alexa Fluor-594) labeled oxLDL (final concentration 5 nM) dissolved in assay buffer were added. The lidded cell plates were incubated for 3h in a 37 °C incubator w/o CO2. After this step, the cell plates were washed 5 times, each time employing 100 μl of wash buffer. 15 μl of wash buffer remained in each well. The plates were measured to determine the amount of AF594- labeled oxLDL, bound to LOX-1. Each assay plate contains wells with Doxycycline induced cells and wells containing non induced cells. Induced cells are the reference for the high signal (100% control), not induced cells are the reference for the low signal (0% control). The fluorescence signal generated is proportional to the amount of AF594 oxLDL bound to LOX-1. The statistical parameters for the screening campaign were as follows: mean value of the Gaussian distribution: 97.2% CTL and SD: 12.5% CTL, Z'-factor 0.74. For the determination of the HTS assay window independent experiments have been performed: +Tet-Alexa oxLDL: $n = 6$; -Tet+Alexa oxLDL: $n = 4$; +Tet+Alexa oxLDL: $n = 6$; 30 nM MAB-LOX, Hycult: $n = 6$. IC50 determinations for BI-0115 are from eight independent measurements. The number of experiments for BI-1580 is $n = 2$ for the LOX-1 assay and $n = 1$ for the SR-B1 assay.

Furthermore, quenching assays were established to eliminate compounds showing quenching of the AF594 or AF647 dye fluorescence. For this, human oxLDL labeled with either AF594 or AF647 was incubated with compound and the fluorescence signal with and without compound was measured to determine the quenching of the fluorescence. For both compounds quenching of fluorescence has been tested with $n = 1$.

**LOX-1 protein production and crystallization**. The cDNA coding for LOX-1 protein amino acid 143-273 and amino acid 129-273 together with a N-terminal 6His-tag followed by a thrombin cleavage site was synthesized at Thermo Fisher Scientific GENEART and cloned into pET28a with NcoI and EcoRI. pET28a containing Lox-1 cDNA was transformed into Shuffle cells (New England BioLabs) that constitutively express a chromosomal copy of the disulfide bond isomerase DsbC. Cells were grown in LB media containing 30 μg/ml kanamycin at 30 °C until an OD600 of 0.6 was reached. Protein expression was induced with 1 mM IPTG and carried out at 16 °C overnight.

Cells were harvested by centrifugation and lysed using the constant cell disruption system (Constant Systems LTD). Purification was performed using Ni-NTA (Qiagen) affinity chromatography, followed by cleavage of the His-tag with thrombin protease and removal of the cleaved His-tag with a second Ni-NTA chromatography. The cleaved LOX-1 protein from the flow-through was further purified by gel filtration chromatography on Superdex 75 (26/60, GE Healthcare) in a buffer containing 20 mM Hepes pH 7,5 and 200 mM NaCl. The protein was concentrated to 3–5 mg/ml for crystallization and 10 % glycerol was added for storage at −80 °C.

LOX-1 143-273 crystallized by the hanging drop vapor diffusion method with a reservoir solution of 30% PEG 2000MME, 0.15 M KBr, pH 6.4–7.1. A 3 mg/ml protein solution was mixed at a ratio of 1.5:1 with the reservoir solution and incubated at 4 °C. For the crystallization of LOX-1 129-273 a reservoir solution containing 0.1 M MES pH 6, 20% PEG6000, 0.2 M LiCl has been mixed 1:1 with the respective protein solution. Apo crystals were soaked overnight by adding 0.1 μL of a 100 mM DMSO stock solution of BI-0115. Co-crystallization was performed with a LOX-1 143-273 protein solution containing 2 mM of the compound. Crystals were obtained with a 1:1 mixture of the protein solution and a reservoir solution containing 0.1 M Bis-Tris pH 7.5, 30% PEG MME 2000 and 0.15 M KBr.

**Data collection, processing and refinement**. Crystals were frozen in liquid nitrogen and data were collected at the SLS beam line X06SA (Swiss Light Source, Paul Scherrer Institute) using the PILATUS 6M detector. LOX-1 143-273 apo crystals belonged to space group P 21 21 21 and contained 2 monomers per asymmetric unit. LOX-1 129-273 apo crystals belonged to space group C2 and contained 3 monomers per asymmetric unit. LOX-1 143-273 BI-0115 crystals belonged to space group C2 and contained 8 monomers per asymmetric unit. Images were processed with APRV[41], XDS[42] and autoPROC[43]. The resolution limits were set using default autoPROC settings. The structures were solved by molecular replacement using MOLREP[44], Phaser[45] and PDB ID 1YPU[24] as a start model. Subsequent model building and refinement was done using standard protocols using CCP4[46] programs, Coot[47], Phenix[48] and autoBUSTER[49]. A summary of the statistics can be found in Supplementary Table 5. Pictures have been made using the PyMOL[50] software.

**Isothermal Titration Calorimetry Analysis**. Isothermal titration calorimetry experiments were conducted on a MicroCal iTC200 instrument (Malvern) using LOX-1 that had been passed through a PD-10 desalting column (GE Healthcare) equilibrated with 20 mM Tris pH 7.3 and 150 mM NaCl. Complete saturation of

35 µM, respectively 40 µM LOX inhibitor was typically achieved by injecting 18× 2 µl aliquots of 300 µM LOX-1 at a temperature of 25 °C. As a control 300 µM LOX-1 was injected in buffer ($n = 1$, Supplementary Fig. 19).

The thermodynamic binding parameters were extracted by non-linear regression analysis of the binding isotherms (MicroCal PEAQ-ITC Analysis Software). A single-site binding model was applied yielding the binding enthalpy (ΔH), stoichiometry (n), entropy (ΔS) and association constant ($K_a$). The binding isotherm of protein in buffer control was not subtracted during data evaluation.

**Surface Plasmon Resonance Analysis.** LOX-1 129-273 was immobilized onto a CM5 chip (GE Healthcare Bio-Sciences AB, Uppsala, Sweden) by amine coupling in 10 mM Na acetate at pH 5.0. Binding studies were performed using a Biacore T200 instrument (GE Healthcare) at 25 °C in 20 mM HEPES pH 6.8, 150 mM NaCl, 0.05% Tween 20, and 1% DMSO at a flow rate of 30 µL/min. The interaction analyses were performed in the multi-cycle kinetic mode using 120 s association time and 600 s dissociation time at the following LOX-1 inhibitor concentrations 0.391, 0.78, 1.56, 3.125, 6.25, 12.5 and 25 µM. Data were analyzed using the Biacore T200 Evaluation Software 2.0. Due to a transient binding, the $K_D$ was determined from a plot of steady state response levels against analyte concentration (affinity plot). The mean $K_D$ value was determined from four individual experiments (Supplementary Table 3). The binding ratio is the average number of compound molecules bound per LOX-1 protein on the surface.

**NMR spectroscopy.** STD NMR binding experiments[51] were performed in 2.5 mm NMR tubes on a Bruker AVII+ 600 MHz spectrometer equipped with a 5 mm cryo-TCI probe and z-gradients. 5 µM LOX129 was incubated with 250 µM BI-0115 in 25 mM Na-phosphate, 100 mM NaCl pH 7.5 in D2O at 298 K and a total d6-DMSO concentration of 1%. The on- and off-resonance spectra were recorded in an interleaved fashion with 256 transients, respectively. Saturation was achieved by irradiation with a Gaussian pulse train for 3 s at resonance frequencies of 100 Hz and 20 kHz, respectively[52]. Residual protein signals were suppressed by employing a 30 ms spin lock pulse prior to acquisition. Water suppression was achieved by a WATERGATE sequence employing a double gradient echo[53]. The signal intensity in the saturation transfer difference spectrum was normalized by the following equation to yield the %STD value: %STD = 100 * (((I(0)-I(sat))/I(0))), where I(0) is the intensity of one signal in the off-resonance and I(sat) the intensity of one signal in the on-resonance spectrum, respectively. Topspin 3.0 (Bruker Biospin) was used to process and analyze data.

**Homology Modelling.** The homology model of the tetrameric complex has been prepared using the MOE[54] software package. The CTLD part is based on the co-crystal structure of the BI-0115-LOX-1 complex, whereas the neck domain is modeled based on the Homer coiled-coil domain PDB ID 3CVE[55]. The Homer coiled-coil sequence showed the highest sequence homology to the LOX-1 neck domain and therefore selected as model basis.

**Mass spectrometry.** Mass spectrometry experiments were carried out on an electrospray time-of-flight mass spectrometer (LCT, Waters, Manchester, UK) equipped with an automated chip-based nanoESI device (Triversa Nanomate, Advion Biosciences, Ithaca, NY). External calibration was performed in positive ion mode over the mass range m/z 500-5000 using the multiply charged ions produced by 0.5 µM horse heart myoglobin solution diluted in water/acetonitrile 50/50 mixture acidified with 0.5% (v/v) formic acid.

Prior to MS analysis, LOX-1 was buffer exchanged against ammonium acetate (AcNH₄) 200mM pH7.5 using 7 cycles of concentration / dilution on size exclusion filter (Amicon®Ultra 0.5 ml, 10kD). Prior to incubation, ligand is dried out from DMSO and solubilized at 1mM in isopranol.

Integrity, homogeneity and purity of LOX-1 was first analyzed under denaturing conditions by diluting the protein to 1 µM in 50/50 water/acetonitrile mixture acidified with 0.5% (v/v) formic acid. Mass measurement revealed the presence of a single species with a molecular weight of 33773.0 ± 0.4Da corresponding to the theoretical molecular weight calculated from a covalent dimer of LOX-1 with 7 disulfide bridges (Supplementary Fig. 9).

Characterization of LOX-1 under non-denaturing conditions was performed in 200 mM NH₄Ac pH 7.5 keeping a constant 5% amount of isopropanol (v/v). Protein concentration was set to 10 µM and compound concentrations tested using a 2.5, 5 and 10 molar excess. Incubations were performed at room temperature for 5 min. Mass spectra were recorded using reduced cone voltage (Vc = 40V) and elevated interface pressure (Pi = 5mbar) which correspond to fine-tuned instrumental settings providing sufficient ion desolvation while preserving the integrity of weak non-covalent complexes in the gas phase. Higher transmission parameter (Vc = 60V) allows better spectrum resolution but induced gas phase dissociation of the ligand. Oligomeric states of LOX-1 are not affected by this tuning and confirm ligand induced LOX-1 tetramerization (Supplementary Fig. 10).

**PAMPA.** The PAMPA assay provides data on the passive permeability of test compounds through immobilized artificial phospholipid membranes. The method used was extensively described previously[56]. The number of experiments for BI-0115 is $n = 1$.

**High Throughput solubility.** The aqueous solubility of the test compound is determined by comparing the amount dissolved in buffer to the amount in an acetonitrile/water (1/1) solution. Starting from a 10 mM DMSO stock solution aliquots are diluted with acetonitrile/water (1/1) or buffer resp. After 24h of shaking, the solutions are filtrated and analyzed by LC-UV. The amount dissolved in buffer is compared to the amount in the acetonitrile solution. Solubility will usually be measured from 0.001 to 0.125 mg/ml at a DMSO concentration of 2.5%. The solubility at the different pH values for BI-0115 were measured with $n = 1$.

**Metabolic stability in microsomes.** The metabolic degradation of the test compound is assayed at 37 °C with microsomes. The final incubation volume of 100 µl per time point contains TRIS buffer pH 7.6 at RT (0.1 M), magnesium chloride (5 mM), microsomal protein (1 mg/ml or 0.5 mg/ml depending on species) and the test compound at a final concentration of 1 µM.

Following a short preincubation period at 37 °C, the reactions were initiated by addition of beta-nicotinamide adenine dinucleotide phosphate, reduced form (NADPH, 1 mM) and terminated by transfering an aliquot into solvent after different time points. The quenched incubations are pelleted by centrifugation (10000 g, 5 min). An aliquot of the supernatant is assayed by LC-MS/MS for the amount of parent compound. The slope of the semi logarithmic plot of the concentration-time profile determines the half-life. The clearance is calculated by upscaling considering the amount of protein in the incubation, microsomal recovery and liver weight and normalized to the hepatic blood flow of the respective species. The number of experiments for BI-0115 is $n = 1$.

**hERG.** hERG-mediated membrane currents are recorded in HEK (human embryonic kidney) 293 cells stably transfected with hERG cDNA, using the whole-cell configuration of the patch-clamp technique as previously described[57]. The number of experiments for BI-0115 is $n = 1$.

**Compound synthesis.** A detailed description of the compound synthesis of BI-0115 and BI-1580 is provided in the Supplementary Methods.

**Reporting summary.** Further information on research design is available in the Nature Research Reporting Summary linked to this article.

## Data availability
The authors declare that the data supporting the findings of this study are available within the paper and its supplementary information files, or from the corresponding author upon reasonable request. The crystallographic data for the LOX-1 extracellular domain in complex with BI-0115 is deposited under accession code 6TL9. The non-liganded LOX143 structure is deposited under accession code 6TL7. The non-liganded LOX129 structure is deposited under accession code 6TLA. Referenced accessions include OLR1_HUMAN, OLR1_MOUSE, OLR1_RAT, Q2HXU8, Q6UXN8, Q9BXN2, 1YPO, 1YPQ, 1YPU, 1YXJ, 1YXK, 3VLG and 3CVE.

## Code availability
For analysis of HTS data, Biolab, a Boehringer Ingelheim proprietary software with restricted access has been used.

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

## Acknowledgements

The authors thank Adelheid Löhle, Yvette Hövels, Katja Mück, Antje Adomeit, Sebastian Eder, Karin Guderlei for expert technical assistance. Christofer Tauterman helped with building the homology model in MOE. Many thanks to Claudia Heine, Katja Phillipp and Anna-Luisa Reiser for providing the chemical synthesis. We thank Dirk Reinert and Expose GmbH, Villigen for data collection and the staff at Swiss Light Source (SLS) beamlines PXI and PXIII for their continuous support. Many thanks to Xavier Espanel and Dominique Roecklin from Novalix for fruitful discussions and clear outputs from the native MS study.

## Author contributions

F.B. conceived the experimental design of the HTS screen, analyzed the data and wrote the manuscript. D.F. conceived the experimental design of the X-ray experiments and contributed to the design of the biophysical experiments, analyzed the data and wrote the manuscript. S.H. analyzed the HTS screening data, performed compound clustering and wrote the manuscript. R.H. conceived the experimental design of fluorescence microscopy experiments, analyzed the data and wrote the manuscript. K.K. supervised in-vitro PK experiments, analyzed the data, and wrote the manuscript. I.L. analyzed SAR data and led the chemistry program. H.Na. contributed to research design, analyzed the data, and wrote the manuscript. H.Ne. contributed to research design and led the biology program, conceived the cloning work and the experimental design of the screening assays, analyzed the data, and wrote the manuscript. J.P. contributed to research design. G.S. conceived the experimental design of the biophysical experiments, analyzed the data, and wrote the manuscript. R.W. contributed to research design and led the chemistry program. M.W. conceived the experimental design of fluorescence microscopy experiments, analyzed the data and wrote the manuscript. M.Z. conceived the experimental design of the STD-NMR experiments, analyzed the data, and wrote the manuscript. F.D. conceived the experimental design of the ESI-MS experiments, analyzed the data, and wrote the manuscript.

## Competing interests

The authors declare the following competing interest: all authors, except F.D., were employees of Boehringer Ingelheim Pharma GmbH & Co KG at the time this study was performed. F.D. is employee of NovAliX.
