## [Peer Review File · Communications Chemistry]

Reviewers' comments:

Reviewer #1 (Remarks to the Author):

In the manuscript of Schnapp et al., "A small-molecule inhibitor of lectin-like oxidized LDL receptor-1 acts by stabilizing an inactive receptor tetramer state" the authors describe the identification of a small molecule (BI-0115) that blocks oxLDL uptake. An in-depth biophysical characterization confirmed binding of BI-0115 to the soluble expressed homodimer LOX-1 via SPR and ITC a K_d value of $\sim 5 \mu\text{M}$. STD-NMR measurements further confirmed receptor binding irrespective of its monomeric or dimeric state. Interestingly, ITC measurements already indicated a surprising monomer to compound binding mode at a 2:1 stoichiometry. BI-0115 induced tetramerization of the CTLD domains was found in co-crystallization studies and non-denaturing MS experiments confirmed this new arrangement. Finally, in a cell based assay oxLDL uptake via LOX-1 was effectively inhibited by BI-0115 (IC_{50} of $\sim 7 \mu\text{M}$). Obviously, kinking of the CTLD in a head-to head topology renders the basic spine ineffective to bind the ligand oxLDL. Further experiments that address potential off-target effects, demonstrate the high selectivity of BI-0115 towards LOX-1.

The authors describe here in detail a novel mechanism how a small molecule that does not directly target the oxLDL receptor binding site inhibits ligand binding in an allosteric manner. This finding is of high interest also for others in this field of research who aim to develop inhibitors addressing related members of the C-type lectin receptors. The work presented here is very convincing, sufficient information is provided to follow, and if desired repeat some experiments. The manuscript is well written and concise. The authors follow a common theme and tell an interesting story. In all parts this is a comprehensible presentation and it was a pleasure to review the manuscript. Needless to say, the relevant literature is up to date.

Nevertheless, I have the following remarks:

- Table 1: please change selectivity to selectivity; LOX-1:LS0115 to LOX-1:BI-0115
- Can you titrate ligand binding if you increase FBS concentration? What about BI-0115 binding to FBS and a potential efficacy in vivo?

I highly recommend to accept the manuscript for publication in Communications Chemistry!

With kind regards,
Jens Dornedde

Charité-Universitätsmedizin Berlin
Augustenburger Platz 1
13353 Berlin

Reviewer #2 (Remarks to the Author):

The paper by Schnapp et al. describes the screening and selection of molecules inhibiting the activity of LOX-1 receptor, using a cell assay. Due to the role of LOX-1 in cardiovascular diseases and cancer, finding specific inhibitors is urgently necessary. Overall, this interesting paper deserves attention.

Authors focused their analysis on a single small molecule and on its in vitro inhibitory mode of action by generating two truncated forms of LOX-1, which contain the extracellular CTLD region and part of the Neck domain of the receptor. Beside a complete biophysical characterization of the interaction, the crystal structure of the compound-LOX-1 complex is provided, which gives atomic details on the interactions and on the formation of a stable and inactive LOX-1 tetramer. From these characterizations, Authors propose that the bound compound, which glues two LOX-1 dimers

together, in a head-to-head mode, stabilizes the formation of inactive tetramers. I have some comments to do on the model proposed.

Comments

1) The new inhibitory mode of action proposed is based on experiments done in vitro with truncated soluble LOX-1 constructs. I am not so convinced that this mechanism can be translated in vivo. The plasma membrane, rich in negative charges, tends to keep the neck domain away and therefore it seems unlikely that the CTLD domains can approach the membrane to form tetramers with head-to-head stable complexes. It should be considered also that LOX-1 is mainly localized in lipid rafts whose characteristic is their great rigidity. Two BI0115 molecules would seem too small to guarantee the formation of stable bonds of very large CTLD domains, considering that they are characterized by positive charges (basic spines) that should repel each other.

If Authors want to give credibility to the proposed model, they must prove it analyzing the BI0115-LOX-1 complexes in cells. For example, in order to experimentally verify whether in living cells BI0115 is involved in the stabilization of LOX-1 receptor tetramers, cells can be treated with the inhibitor or the natural substrate ox-LDL and then the tetramers (or dimers or octamers) vs. monomer ratio can be evaluated either purifying full-length LOX-1 receptors from cell membranes or collecting the soluble form of LOX-1 from conditioned medium. In this way it has been proved that statins, by filling the hydrophobic tunnel, stabilize a dimeric form of LOX-1 in human cells (Cell Cycle, 14, 1583-95 2015).

2) In the Introduction it is stated: "A hydrophobic tunnel at the homodimer interface has been observed. ... The function of this tunnel remains unclear." This last sentence is debatable. Authors mention some papers in the Discussion that, using site directed mutagenesis, or molecular dynamic simulations and experimentally, answer to this question demonstrating the role of the hydrophobic tunnel in the recognition of ox-LDL.

Reviewer #3 (Remarks to the Author):

In their report, Fiegen and coworkers, report on the identification of a novel low molecular weight inhibitor of LOX-1 and its biophysical characterization. They applied a large repertoire of methods combining cell-based HTS, SPR, NMR, ITC, X-ray and chemistry to provide a high affinity and specific chemical probe to the community. Besides providing insights into a well organized and streamlined development platform, with emphasis on the importance of biophysical methods and their advantage in drug discovery, they present one of the rare examples of a drug-like molecule inhibiting a member of a challenging target class. On top, a novel mode of action (4:2 clustering) has been thoroughly described. Overall, I recommend the publication of this work with minor revisions.

Minor comments:

- Applied to cells, does BI-0115 induce downstream signaling, and does it lead to larger agglomerates of LOX-1 receptors on the cell surface e.g. in such a way that a (4+2) cluster engages another cluster?
- Were other C-type lectins tested as off-targets and does the binding site and mode make this a likely scenario?
- please change „NMR was used to prove ...“ since proof is a very strong word in this context. May be showed or further supported ...
- 18.5% STD, does this refer to 18.5% STD amplification factor? Is this a normalized value?
- In the text the Kd is given as 7uM, in the figure caption as 4.3 uM. Please be consistent.
- Figure 2 (a) please show also the aliphatic resonances to exclude direct saturation of the ligand.

Comment: I believe you can get rid of the baseline artifacts, assuming a water suppression was used (please include in the methods) adjusting the lock phase. (b) Please plot the data points in the fit, this is barely readable. Which model was used to fit the data? (c) Was a protein in buffer injection subtracted from the data?

- „10 Kelvin“ should read „10 K“ or better 10°C since this easier to understand for the general audience.

- In the SPR Methods section, a kinetic analysis is described while the fits in figure 2 look like equilibrium fits. Please double check the exact procedure.

-Figure S9: are these data sequence identities or similarities?

Letter to reviewers

Reviewers' comments:

Reviewer #1 (Remarks to the Author):

In the manuscript of Schnapp et al., "A small-molecule inhibitor of lectin-like oxidized LDL receptor-1 acts by stabilizing an inactive receptor tetramer state" the authors describe the identification of a small molecule (BI-0115) that blocks oxLDL uptake. An in-depth biophysical characterization confirmed binding of BI-0115 to the soluble expressed homodimer LOX-1 via SPR and ITC a K_d value of $\sim 5 \mu\text{M}$. STD-NMR measurements further confirmed receptor binding irrespective of its monomeric or dimeric state. Interestingly, ITC measurements already indicated a surprising monomer to compound binding mode at a 2:1 stoichiometry. BI-0115 induced tetramerization of the CTLD domains was found in co-crystallization studies and non-denaturing MS experiments confirmed this new arrangement. Finally, in a cell based assay oxLDL uptake via LOX-1 was effectively inhibited by BI-0115 (IC_{50} of $\sim 7 \mu\text{M}$). Obviously, kinking of the CTLD in a head-to head topology renders the basic spine ineffective to bind the ligand oxLDL. Further experiments that address potential off-target effects, demonstrate the high selectivity of BI-0115 towards LOX-1.

The authors describe here in detail a novel mechanism how a small molecule that does not directly target the oxLDL receptor binding site inhibits ligand binding in an allosteric manner. This finding is of high interest also for others in this field of research who aim to develop inhibitors addressing related members of the C-type lectin receptors. The work presented here is very convincing, sufficient information is provided to follow, and if desired repeat some experiments. The manuscript is well written and concise. The authors follow a common theme and tell an interesting story. In all parts this is a comprehensible presentation and it was a pleasure to review the manuscript. Needless to say, the relevant literature is up to date.

A: We thank the reviewer very much for these supportive comments.

Nevertheless, I have the following remarks:

-Table 1: please change seletivity to selectivity; LOX-1:LS0115 to LOX-1:BI-0115

A: The table was corrected accordingly.

-Can you titrate ligand binding if you increase FBS concentration? What about BI-0115 binding to FBS and a potential efficacy in vivo?

A: A titration experiment of ligand with increasing FCS concentration was not performed. For this compound series, a serum shift assay was not performed and binding of BI-0115 to FCS was not tested. Based on the available data we cannot draw conclusions on the in vivo activity of the compounds. The low sequence conservation between human and rodents make BI-0115 an ideal probe for studying human in vitro or humanized in vivo systems. Performing additional cellular experiments would mean a large effort due to the labor-intensive preparation of fluorescence labeled human oxLDL.

I highly recommend to accept the manuscript for publication in Communications Chemistry!

With kind regards,

Jens Dornedde

Charité-Universitätsmedizin Berlin

Augustenburger Platz 1

13353 Berlin

Reviewer #2 (Remarks to the Author):

The paper by Schnapp et al. describes the screening and selection of molecules inhibiting the activity of LOX-1 receptor, using a cell assay. Due to the role of LOX-1 in cardiovascular diseases and cancer, finding specific inhibitors is urgently necessary. Overall, this interesting paper deserves attention. Authors focused their analysis on a single small molecule and on its in vitro inhibitory mode of action by generating two truncated forms of LOX-1, which contain the extracellular CTLD region and part of the Neck domain of the receptor. Beside a complete biophysical characterization of the interaction, the crystal structure of the compound-LOX-1 complex is provided, which gives atomic details on the interactions and on the formation of a stable and inactive LOX-1 tetramer. From these characterizations, Authors propose that the bound compound, which glues two LOX-1 dimers together, in a head-to-head mode, stabilizes the formation of inactive tetramers. I have some comments to do on the model proposed.

A: We thank the reviewer for these supportive comments.

C: The presented study focuses on BI-0115 and the corresponding control compound. The data on the two highlighted compounds is confirmed by a series of similar compounds in the Structure activity relationship (SAR) overview (Supplementary figure 6). The tetramerization model fits and explains the observed SAR. The confined space in the tetramer allows only very small changes.

Comments

1) The new inhibitory mode of action proposed is based on experiments done in vitro with truncated soluble LOX-1 constructs. I am not so convinced that this mechanism can be translated in vivo.

A: The reviewer correctly spotted that for the biophysical studies we used truncated soluble forms of LOX-1 to confirm target binding and identify the BI-0115 interaction site and its mode-of-action. However, the cellular HTS screen was done using full-length LOX-1 and human oxLDL (CHO-Trex-hLOX1 cell line, see methods section) and the IC₅₀ determinations were conducted with the same assay setup. These results clearly show that the compound is active on full-length LOX-1 in the cellular context. The functional in vitro data, namely the blocking of human oxLDL uptake, in our view strongly suggest that the mode-of-action should translate also to the in vivo context.

The plasma membrane, rich in negative charges, tends to keep the neck domain away and therefore it seems unlikely that the CTLD domains can approach the membrane to form tetramers with head-to-head stable complexes. It should be considered also that LOX-1 is mainly localized in lipid rafts whose characteristic is their great rigidity. Two BI0115 molecules would seem too small to guarantee the formation of stable bonds of very large CTLD domains, considering that they are characterized by positive charges (basic spines) that should repel each other.

A: Our biophysical and structural data clearly show that despite the positively charged basic spine surface feature, facilitated by enclosing two BI-0115 molecules, two LOX-1 CTLD domains are perfectly able to bind to each other. Concerning the approach of two lipid raft membrane bound LOX-1 receptors we do not see a problem. Since the NECK domain itself and especially the kink between NECK and CTLD domains are flexible, it is not necessary to invoke changes in the distance of the CTLDs to the membrane nor in the plasticity of the lipid raft to support our inactivation model. LOX-1 extracellular domain flexibility is indeed a crucial part for its functionality and LOX-1 dimer association to oligomers facilitates binding of differently sized ligands (Cao et al. 2009).

*Regarding the small size of the compound, the example of Brefeldin A (molecular weight of 280 g/mol) in the stabilization of the ARF1*Sec7 complex proves that small molecules similar in size to BI-0115 can stabilize the interaction between proteins (Mossessova et al. 2003 Mol Cell). Additionally, not only the two BI-0115 ligands contribute to the interaction between the two dimers but also the LOX-1 protein. In our opinion, it is not the size of the compound that is important for its potency, but the quality of its interactions (ligand efficiency of 0.32).*

If Authors want to give credibility to the proposed model, they must prove it analyzing the BI0115-LOX-1 complexes in cells. For example, in order to experimentally verify whether in living cells BI0115 is involved in the stabilization of LOX-1 receptor tetramers, cells can be treated with the inhibitor or the natural substrate ox-LDL and then the tetramers (or dimers or octamers) vs. monomer ratio can be evaluated either purifying full-length LOX-1 receptors from cell membranes or collecting the soluble form of LOX-1 from conditioned medium. In this way it has been proved that statins, by filling the hydrophobic tunnel, stabilize a dimeric form of LOX-1 in human cells (Cell Cycle, 14, 1583-95 2015).

A: As discussed above, we unambiguously show that BI-0115 works in the cellular context and inhibits the uptake of human oxLDL, much like the data provided in the paper that the reviewer refers to. Further,

in contrast to the mentioned literature, we have experimental evidence from multiple different technologies for our model, whereas Biocca et al. rely on in-silico studies and an artificial (extremely high reducing agent) dimer stabilization assay to propose a mode-of-action for the statins. This model suggests a “stabilization” of the receptor dimer (that already exists as the functional form of the receptor). Furthermore, the statin concentrations used in Biocca et al. are in the 2-10 μ M range, far from typical C_{max} concentrations of statins in human serum (6-50 nmol/L; Björkhem-Bergman et al. 2011 BJCP). Whereas the statins show only a partial inhibition of oxLDL uptake, we can show with BI-0115 a full inhibition in the cellular context.

We strongly believe that the volume of our data is sufficient to warrant publication without the proposed additional cellular validation. We think that additional experiments to prove the tetrameric state in cells is not needed in the scope of this publication, but would certainly be interesting experiments to conduct. In order to facilitate work in this direction, we make BI-0115 publicly available. We think that the proposed assay (ultra centrifugation from cell lysates) does not provide the resolution to discriminate between dimers and tetramers.

The sentence “For none of these compounds there is any direct evidence for their mode of action.” was removed from the discussion.

2) In the Introduction it is stated: "A hydrophobic tunnel at the homodimer interface has been observed. ... The function of this tunnel remains unclear." This last sentence is debatable. Authors mention some papers in the Discussion that, using site directed mutagenesis, or molecular dynamic simulations and experimentally, answer to this question demonstrating the role of the hydrophobic tunnel in the recognition of ox-LDL.

A: The respective sentence has been deleted and replaced by “Nakano et al.²⁶ could show that residues surrounding this tunnel are important in the self-assembly of the canonical dimer.”

Reviewer #3 (Remarks to the Author):

In their report, Fiegen and coworkers, report on the identification of a novel low molecular weight inhibitor of LOX-1 and its biophysical characterization. They applied a large repertoire of methods combining cell-based HTS, SPR, NMR, ITC, X-ray and chemistry to provide a high affinity and specific chemical probe to the community. Besides providing insights into a well organized and streamlined development platform, with emphasis on the importance of biophysical methods and their advantage in drug discovery, they present one of the rare examples of a drug-like molecule inhibiting a member of a challenging target class. On top, a novel mode of action (4:2 clustering) has been thoroughly described. Overall, I recommend the publication of this work with minor revisions.

A: We thank the reviewer very much for these supportive comments.

Minor comments:

- Applied to cells, does BI-0115 induce downstream signaling, and does it lead to larger agglomerates of LOX-1 receptors on the cell surface e.g. in such a way that a (4+2) cluster engages another cluster?

A: In the cellular context, we could show that the BI-0115 compound is active on full-length LOX-1 and inhibits uptake of human oxLDL. The inhibition of downstream signaling by BI-0115 was not investigated but we interpret the blocking of the oxLDL uptake as a sign for inhibition of downstream signal transduction.

We have not performed any experiments that would elucidate further oligomerisation of the LOX-1 tetramers. Based on the structural information, we would presume that this would not be a preferred process.

- Were other C-type lectins tested as off-targets and does the binding site and mode make this a likely scenario?

A: A very good comment. Based on the low sequence conservation between C-type lectins it is unlikely that BI-0115 is going to bind to other isoforms and we therefore have not tested any other C-type lectins as off-targets. The closest paralogues are CLEC7A, CLEC9A and CLEC12B with a sequence identity below 40%. For illustration, we added two figures to the supplement, including a sequence alignment. This shows that the residues around the BI-0115 binding site are not conserved between these closest paralogues. A corresponding paragraph was added to the discussion.

- please change „NMR was used to prove ...“ since proof is a very strong word in this context. May be showed or further supported ...

A: The respective paragraph was changed.

As a first method, Saturation Transfer Difference (STD)-NMR was used to demonstrate binding to the receptor.

- 18.5% STD, does this refer to 18.5% STD amplification factor? Is this a normalized value?

*A: The %STD value is the intensity of the difference spectrum normalized by the internal reference spectrum (off resonance) spectrum. We added the following equation to the Methods section for clarification $\%STD = 100 * ((I(0) - I(sat)) / I(0))$ and describe its use in more detail.*

- In the text the Kd is given as 7uM, in the figure caption as 4.3 uM. Please be consistent.

A: The respective value in the figure caption was changed.

- Figure 2 (a) please show also the aliphatic resonances to exclude direct saturation of the ligand. Comment: I believe you can get rid of the baseline artifacts, assuming a water suppression was used (please include in the methods) adjusting the lock phase.

A: We do not believe that showing the aliphatic region adds clarity to the figure since this region of the spectrum is dominated by the very intense DMSO signal and residual HEPES signals from the protein

buffer. In addition, we can exclude direct saturation of the ligand because the methyl signal of the ligand at 0.9 ppm shows a comparable %STD value as the resonances in the depicted aromatic region. This statement is now also included in the main text.

Regarding water suppression: We used a WATERGATE water suppression scheme, which we now also state in the respective Methods section and included the references.

(b) Please plot the data points in the fit, this is barely readable. Which model was used to fit the data?

The data points in the fit were enlarged to enable better readability.

The used model calculates the equilibrium dissociation constant K_D for a 1:1 interaction from a plot of steady state binding levels (R_{eq}) against analyte concentration (C).

(c) Was a protein in buffer injection subtracted from the data?

The protein in buffer injection was performed as a control (is included as Supplement figure 17). Since the protein in buffer control did not show a change of heat generation, it was not subtracted from the data.

- „10 Kelvin“ should read „10 K“ or better 10°C since this easier to understand for the general audience.

A: The respective sentence was changed.

- In the SPR Methods section, a kinetic analysis is described while the fits in figure 2 look like equilibrium fits. Please double check the exact procedure.

A: The respective paragraph was changed.

-Figure S9: are these data sequence identities or similarities?

A: The percent values in the figure are sequence identities. The figure caption was changed: “Supplement Figure 9: Sequence conservation across human, rat and mouse. The given percent values reflect sequence identities for the full-length proteins.”

REVIEWERS' COMMENTS:

Reviewer #1 (Remarks to the Author):

The authors provide a valid revision of the manuscript and replied appropriate to all remarks of the reviewers.

The manuscript should be published in Communications Chemistry.

Reviewer #2 (Remarks to the Author):

Comments to the revised version.

I understand the point of view of the Authors that "performing additional cellular experiments would mean a large effort ..." and I agree that further cell experiments are out of the scope of this publication. Thus, I am satisfied with the changes made in the revised version.

Reviewer #3 (Remarks to the Author):

All comments have been addressed.